# Association between Hematological Indicesand Disease Activity in Patients with Rheumatoid Arthritis Treated with Janus Kinase Inhibitors for 24 Weeks

**DOI:** 10.3390/medicina58030426

**Published:** 2022-03-15

**Authors:** Jung-Yoon Choe, Seong-Kyu Kim

**Affiliations:** Division of Rheumatology, Department of Internal Medicine, Catholic University of Daegu School of Medicine, Daegu 42472, Korea; jychoe@cu.ac.kr

**Keywords:** rheumatoid arthritis, JAK inhibitor, hematological index, disease activity, DAS28

## Abstract

*Background and Objective*: Hematological indices have been considered reliable markers for assessment of disease activity in rheumatoid arthritis (RA). This study assessed whether hematological indices reflect changes in disease activity in patients with RA treated with Janus kinase (JAK) inhibitors. *Materials and Methods*: This study recruited 123 patients with RA who completed a regimen of JAK inhibitors, including baricitinib or tofacitinib, for 24 weeks, and 80 age- and sex-matched healthy control subjects. Hematological indices were systemic immune-inflammation index (SII), neutrophil-to-hemoglobin and lymphocyte (NHL) score, neutrophil-to-lymphocyte ratio (NLR), and platelet-to-lymphocyte ratio (PLR). Disease Activity Score 28 joints using erythrocyte sedimentation rate (DAS28-ESR) was evaluated as a measure of RA disease activity. *Results*: At baseline, patients with RA had a significantly higher SII, NHL score, NLR, and PLR than controls (*p* < 0.001 for all). SII, NHL score, NLR, and PLR at baseline were associated with DAS28-ESR (*p* < 0.05 for all). Changes in SII, NHL score, NLR, and PLR were associated with those in DAS28-ESR during treatment with JAK inhibitors. Such treatment markedly decreased SII, NHL score, and NLR values compared to those at baseline (*p* < 0.001 for all) but did not decrease PLR (*p* = 0.056). There were no differences in changes in SII, NHL score, NLR, and PLR between baricitinib and tofacitinib treatments. No hematological index showed predictive potential with respect to non-response to JAK inhibitor treatment. *Conclusions*: This study showed that hematological indices might be useful in monitoring changes in disease activity in patients with RA treated with JAK inhibitors.

## 1. Introduction

Rheumatoid arthritis (RA) is a systemic autoimmune disease characterized by enhanced synovial inflammation, neovascularization, and osteoclast formation, causing inflammatory arthritis and progressive joint damage that leads to deterioration of quality of life and increased functional disability [1,2,3,4]. Although the pathogenesis of RA has not been clearly defined, there have been dramatic changes in the treatment of RA over the past two decades based on significant advances in the understanding of its pathogenic mechanism. It is important to understand the clinical and radiographic treatment goals, adverse effects of therapeutic agents, disease activity measurement, and assessment of treatment response for optimal treatment of RA.

Numerous measurement tools assessing and monitoring RA disease activity are available in clinical practice [5]. Among them, composite indices, such as Disease Activity Score 28 joints (DAS28) [6], Simplified Disease Activity Index [7], and Clinical Disease Activity Index [8] have been recommended to calculate the treatment response of RA and are used most commonly and feasibly in clinical trials [9]. However, efforts continue to explore easier and more useful markers to reflect treatment response or disease activity. Hematological indices, such as neutrophil-to-lymphocyte ratio (NLR) and platelet-to-lymphocyte ratio (PLR), have attracted attention to novel indicators reflecting disease activity in RA [10,11,12].

For appropriate control of significant inflammation and damage of joints in RA, conventional disease-modifying anti-rheumatic drugs (cDMARDs) have been the standard therapeutics for RA, consisting of methotrexate alone or in combination with other cDMARDs [1]. The advent of biological disease-modifying anti-rheumatic drugs (bDMARDs), such as tumor necrosis factor-α (TNF-α) inhibitors, interleukin-6 receptor blockers, and B cell depletion agents, which are differentiated from cDMARDs, has been a breakthrough in the treatment of RA. Recently, evidence that hematological indices are reliable markers of disease activity even in patients treated with these bDMARDs has been recognized [13,14,15,16]. In addition, Janus kinase (JAK) inhibitors, now classified as targeted synthetic DMARDs (tsDMARDs), are a novel therapeutic option available to patients with RA [17,18]. Small-molecule JAK inhibitors are oral targeted DMARDs, characterized by the blockade of the intracellular JAK–STAT pathway mediated by multiple cytokines involved in the immune-mediated inflammatory response in RA, unlike bDMARDs that regulate inflammatory cytokines, T-lymphocytes or B-lymphocytes. No studies have evaluated whether hematological indices more accurately reflect the disease activity of patients treated with JAK inhibitors compared to traditional disease activity indicators. The purpose of this study is to investigate whether hematological indices, such as NLR and PLR, reflect changes in disease activity in patients with RA treated with JAK inhibitors, including tofacitinib and baricitinib, for 24 weeks.

## 2. Materials and Methods

### 2.1. Study Population

This study retrospectively enrolled 123 patients with RA and 80 healthy age- and sex-matched controls from January 2016 to August 2021. RA patients met the classification criteria for RA as proposed by the American College of Rheumatology (ACR)/European League Against Rheumatism (EULAR) in 2010 [19]. We initially reviewed the medical records of patients with RA treated with JAK inhibitors (55 patients with tofacitinib and 68 patients with baricitinib) due to lack of adequate response to or intolerance of conventional DMARDs, including methotrexate, leflunomide, sulfasalazine, hydroxychloroquine, azathioprine, or tacrolimus and glucocorticoid in outpatient clinics (Figure 1). Among them, 115 RA patients (51 patients with tofacitinib and 64 patients with baricitinib) completed treatment with JAK inhibitors for 24 weeks. Eight patients were dropped due to adverse effects (*n* = 6) or follow-up loss (*n* = 2).

### 2.2. Collection of Clinical Information

Clinical information was collected from patients with RA through a review of medical records and individual interviews with participants at the time of initial JAK inhibitor administration. Characteristics at baseline of both RA patients and controls included age (years), sex, disease duration (months), rheumatoid factor (RF, IU/mL), anti-cyclic citrullinated peptide (anti-CCP) antibody (U/mL), erythrocyte sedimentation rate (ESR, mm/h), and C-reactive protein level (CRP, mg/L). For assessment of disease activity parameters in RA, ESR, CRP, tender joint count, swollen joint count, patient and physician global assessment using a visual analogue scale (VAS), and DAS28-ESR were assessed. The concomitant glucocorticoids and cDMARDs, including methotrexate, hydroxychloroquine, sulfasalazine, leflunomide, and tacrolimus, in use when JAK inhibitor treatment was initiated, were identified.

### 2.3. Assessment of Hematological Indices

We measured white blood cell (WBC), hemoglobin, platelet, neutrophil (%), and lymphocyte (%) counts both before and after use of JAK inhibitors for 24 weeks. Absolute neutrophil and lymphocyte counts were calculated by multiplying the WBC count by the fraction of neutrophils and lymphocytes, respectively. Hematological indices were platelet x neutrophil-to-lymphocyte ratio for systemic immune-inflammation index (SII), neutrophil-to-(hemoglobin x lymphocyte) (NHL) score, NLR, and PLR. The unit for NHL score was 1/g/dL.

### 2.4. Statistical Analysis

Data were described as means and standard deviations (SD) or medians with interquartile ranges (IQR) for quantitative variables and numbers with percentages for qualitative variables. The normality of the data was verified using the Kolmogorov–Smirnov test. Hematological indices were not normally distributed. Comparisons of differences in variables between two groups (patients with RA vs. controls or baricitinib group vs. tofacitinib group) were performed using Student’s *t*-tests, Mann–Whitney U test, or chi-squared tests. Correlations between changes in hematological indices and changes in disease activity parameters during treatment with JAK inhibitors were measured using Pearson correlation analysis. Differences in hematological indices and disease activity parameters between baseline and after 24 weeks of treatment with JAK inhibitors were compared using a paired samples *t*-test or the Wilcoxon signed-rank test.

The European League Against Rheumatism (EULAR) response criteria based on DAS28 was identified as good, moderate, or no response. Patients who showed good and moderate responses were defined as the response group. In the analysis for non-response to treatment with JAK inhibitors for 24 weeks, univariate logistic regression was performed to determine the variables that showed a statistically significant difference between the response group and the non-response group. For significant variables in univariate logistic regression, multivariate regression analysis, including model 1 and model 2, was performed with adjustment for age, sex, disease duration, or DAS28-ESR. The data were described as odds ratios (ORs) and 95% confidence intervals (CIs). Statistical analyses were performed using SPSS version 19.0 (SPSS Inc., Chicago, IL, USA). A *p*-value less than 0.05 was considered statistically significant.

## 3. Results

### 3.1. Baseline Characteristics of the Study Population

There were 123 patients with RA treated with JAK inhibitors and 80 healthy sex- and age-matched controls enrolled initially in the study (Table 1). Among RA patients, 55 (44.7%) were treated with tofacitinib, and 68 (55.3%) were treated with baricitinib.

The mean disease duration of patients with RA was 189.0 months (SD 87.4 months) at the time of initial treatment with JAK inhibitors. RF titer, anti-CCP titer, and acute phase reactants, including ESR and CRP, were significantly different between RA patients and controls (*p* < 0.001 for all). The mean DAS28-ESR in patients with RA was 6.2 (SD 0.7), which indicates high disease activity. Medication use at the time of starting JAK inhibitors was identified, and more than 90% of patients were being treated with glucocorticoids or methotrexate.

### 3.2. Comparison of Hematological Indices between RA Patients and Controls

Patients with RA showed higher WBC counts, platelet counts, proportions of neutrophils (%), and absolute counts of neutrophils than did the controls (*p* < 0.001 for all) (Table 2). In contrast, lower hemoglobin levels, lymphocyte proportions (%), and absolute counts of lymphocytes were found in RA patients compared to controls (*p* < 0.001, *p* < 0.001, and *p* = 0.027, respectively). In a comparison of hematological indices, RA patients had higher SII, NHL scores, NLR, and PLR than did the controls (*p* < 0.001 for all).

### 3.3. Correlation of Hematological Indices with Disease Activity Parameters

In the analysis of the associations between DAS28-ESR and hematological indices at baseline, DAS28-ESR was significantly associated with SII (β = 0.311, *p* < 0.001), NHL score (β = 0.256, *p* = 0.005), NLR (β = 0.202, *p* = 0.027), and PLR (β = 0.266, *p* = 0.003) (Figure 2). We compared the correlations between changes in hematological indices and disease activity parameters in 115 RA patients who completed JAK inhibitor treatment for 24 weeks (Table 3). Changes in all hematological indices, such as SII, NHL score, NLR, and PLR, for 24 weeks were significantly associated with changes in DAS28-ESR and ESR. Changes in CRP and physician VAS for 24 weeks were associated with those in SII and PLR (*r* = 0.208, *p* = 0.026 and *r* = 0.272, *p* = 0.039, respectively). However, tender joint count, swollen joint count, and patient VAS were not correlated with any hematological indices.

### 3.4. Comparison of Changes in Disease Activity Markers before and after JAK Inhibitor Treatment

DAS28-ESR, SII, NHL score, NLR, ESR, CRP, proportion of neutrophils (%), and absolute count of neutrophils at the time of pretreatment with JAK inhibitors were significantly lower than the corresponding post-treatment values (*p* < 0.001 for all) (Table 4). In addition, patients with JAK inhibitor treatment showed increased proportions of lymphocytes (%) and absolute counts of lymphocytes compared to those without JAK inhibitors (*p* < 0.001 and *p* = 0.007, respectively). In contrast, there was no difference in PLR between patients with and without JAK inhibitor use (*p* = 0.056).

### 3.5. Analysis of Variables Related to Non-Response to JAK Inhibitors

This study found 8 patients with good response, 96 patients with moderate response, and 11 patients with no response based on EULAR response criteria among the 115 patients who completed JAK inhibitor treatment for 24 weeks. We found no differences in hematological indices of SII, NHL score, NLR, and PLR at baseline between the response group, which included patients with good and moderate response, and the no-response group (Appendix A). In an assessment of predictive variables for non-response after use of JAK inhibitors for 24 weeks, univariate analysis showed that swollen joint count, RF titer, and CRP were related to non-response (Table 5). None of the hematological indices investigated were associated with non-response after treatment with JAK inhibitors. Model 1 analysis after adjustment for age and sex showed that swollen joint count was a predictive variable for non-response to JAK inhibitors (OR 1.194, 95% CI 1.018–1.400, *p* = 0.030). Model 2 analysis after adjustment for confounding variables also showed that swollen joint count was associated with non-response (OR 1.250, 95% CI 1.027–1.521, *p* = 0.026).

## 4. Discussion

JAK inhibitors, small molecular compounds, and a novel class of oral tsDMARDs are characterized by blockage of intracellular signaling along the JAK–STAT pathway, which plays a crucial role in regulation of the inflammatory response in inflammatory or immune cells [17,18]. Some oral JAK inhibitors, such as tofacitinib, baricitinib, and upadacitinib, are widely available for treatment of RA and other immune-mediated diseases [20]. Tofacitinib and baricitinib are tsDMARDs that have garnered increasing interest, having proven a therapeutic effect equivalent to that of other bDMARDs through clinical studies [17,18]. In most of these studies, DAS28-ESR or DAS28-CRP was used to evaluate the therapeutic efficacy of JAK inhibitors. Thus, the purpose of this study was to verify the usefulness of the hematological indices, including SII, NHL score, NLR, and PLR, to evaluate the therapeutic effect of JAK inhibitors, such as tofacitinib and baricitinib, in patients with RA. The main finding of this study was that SII, NHL score, NLR, and PLR are related closely to DAS28-ESR, and changes in these hematological indices before and after use of JAK inhibitors were significantly associated with those in DAS28-ESR. However, none of the hematological indices had predictive potential for non-response after JAK inhibitor treatment in RA.

Hematological indices, such as NLR, PLR, mean platelet volume, and red cell distribution width, have emerged as inflammatory indicators based on changes in erythrocyte, neutrophil, lymphocyte, or platelet counts or volume of inflammatory conditions, including neoplasms, prothrombotic diseases, or coronary artery disease [21,22,23]. In addition, evidence has shown that hematological indices, such as NLR and PLR, are closely related to disease activity parameters in patients with RA [10,11,12,13,14,15,16]. Cross-sectional clinical studies confirmed that patients with RA had higher NLR or PLR than controls, and hematological indices have been found to be highly correlated with disease activity parameters, including ESR, CRP, and DAS28-ESR [10,11,12]. Uslu et al. showed that NLR and PLR in remission were markedly lower than in active disease, including low, moderate, and severe disease activity based on DAS28 [10]. Similarly, as the disease activity of DAS28 worsened from remission to high status, NLR was increased [11]. In addition to NLR and PLR, mean platelet volume, another hematological index, was lower than the controls and was inversely related to composite index DAS28 in RA [24,25]. Comparably, our study also found that four hematological indices at baseline were significantly associated with DAS28-ESR, ESR, and CRP. In contrast, Yolbas et al. revealed no difference of PLR and NLR between active and quiescent RA patients [26]. Nevertheless, these hematological indices are significantly increased in inflammatory conditions, such as RA.

Several studies have recognized that hematological indices reflect changes in disease activity relatively well before and after treatment with cDMARDs and/or bDMARDs for RA. Maden et al. showed that NLR value was significantly decreased together with ESR and CRP in patients with RA, but another index, mean platelet volume, remained unchanged after treatment [24]. A retrospective study that analyzed 82 female RA patients treated with TNF-α inhibitors for 12 weeks demonstrated that high NLR and PLR subgroups at baseline showed an increased rate of EULAR non-response compared to low NLR and PLR subgroups (*p* = 0.01 and *p* = 0.047, respectively) and that these were significantly associated with an increased risk of EULAR non-response [13]. In addition, patients with high NLR were linked to increased withdrawal of TNF-α inhibitors due to a lack of efficacy. However, the changes in DAS28-ESR at 12 weeks were not significantly different between the high and low NLR and PLR groups at baseline. A different study showed that NLR and PLR values at pretreatment were significantly lower than those at 6 months post-treatment with rituximab [14]. Another retrospective analysis of 52 RA patients revealed that NLR at preflare time was significantly increased compared to that at flare during tocilizumab treatment in patients with RA, but PLR was not changed between the two time points [15]. In addition, an analysis of 358 RA patients treated with numerous bDMARDs, including TNF-α inhibitors, tocilizumab, and abatacept, found that change in NLR was related modestly to change in DAS28-ESR [16]. In our study, we found a significant association between changes in the hematological indices of SII, NHL score, NLR, PLR, and DAS28-ESR after treatment with JAK inhibitors. This suggests that hematological indices are significantly correlated with composite indices, such as DAS28-ESR, and might be indicators that reflect well the degree of inflammatory response in RA.

Although hematological markers might be reliable compared to traditional disease activity indices, there is one thing to consider when they are regarded as an inflammatory index for patients treated with JAK inhibitors. Unlike other bDMARDs, such as tocilizumab, rituximab, and TNF-α inhibitors, changes in counts of platelets, neutrophils, hemoglobin, or lymphocytes appear in the early period during treatment with JAK inhibitors. JAK2 is an essential signaling molecule involved in thrombopoietin- and erythropoietin-mediated hematopoiesis [27,28]. In addition, the JAK signaling system mediates the signaling of cytokines involved in the function of lymphocytes [29]. Baricitinib, an oral JAK1 and JAK2 inhibitor, might be related to transient increased absolute platelet and lymphocyte counts and decreased neutrophil counts, which usually return toward baseline levels within 2 to 24 weeks [30,31]. In addition, hemoglobin level decreased transiently and then returned to a normal range during baricitinib treatment. It is presumed that production of erythrocytes is stimulated through the JAK2 signaling pathway. Another JAK inhibitor, tofacitinib, is a non-selective JAK inhibitor that preferentially blocks the JAK1, JAK2, and JAK3 signaling pathways. An analysis of six randomized controlled trials found that tofacitinib treatment induced a decrease in mean absolute neutrophil and lymphocyte counts [32]. However, neutrophil count usually was stabilized within 3 months. Hemoglobin level has been noted to increase gradually after tofacitinib treatment [32]. Changes in hematological parameters during treatment with JAK inhibitors tofacitinib or baricitinib were considered minimal and returned to baseline or stabilized over time. In addition, less than 0.1–1% of patients treated with JAK inhibitors had to discontinue treatment due to adverse hematological effects [30]. Furthermore, these hematological changes are thought to be the effects of multifactorial causes other than JAK inhibitors, as the increased platelet counts and hemoglobin levels that are involved theoretically in the JAK2 pathway might be considered paradoxical phenomena. Due to the differing degrees of selective inhibition of JAK2 between JAK inhibitors, including tofacitinib and baricitinib, there is a possibility that hematological indices between JAK inhibitors are different. In our study, we tried to verify whether there were differences in changes in hematological indices, neutrophil, lymphocyte, and platelet counts, and hemoglobin levels between tofacitinib and baricitinib treatment for 24 weeks, and there were no significant differences in these markers except platelet count between the two treatments (Appendix A). This suggests that hematological indices are not significantly affected by changes in blood cells during treatment.

This study could not determine the predictive value of hematological indices at baseline on non-response using the EULAR response criteria after treatment with JAK inhibitors for 24 weeks. All hematological indices at baseline in patients with non-response at 24 weeks after JAK inhibitor treatment were similar to those in the response group. Comparable with our results, changes in NLR did not predict the response to bDMARDs in Japanese RA patients, although they reflected the efficacy of bDMARDs treatment based on EULAR response [16]. Another study found that baseline NLR and PLR were not different between stable and flare groups in patients treated with tocilizumab [15]. This suggests that hematological indices, including NLR and PLR, lack the potential to predict clinical outcomes during anti-inflammatory treatment. On the contrary, Lee et al. found that high baseline NLR and PLR were associated independently with increased risk of non-response in patients treated with TNF-α inhibitors for 12 weeks [13]. However, these indices did not show a predictive effect on non-response at 24 weeks. Since there are still debates on the predictive value of hematological indices on treatment response, additional research is needed.

Diverse RA-related disease activity composite indices have been developed, including DAS28, the simplified disease activity index (SDAI), and the clinical disease activity index (CDAI), to use in clinical practice and in research [5,6]. In particular, CDAI and SDAI are currently considered more reliable and comprehensive tools for RA disease activity measurement and prediction of radiographic change. However, most studies verified the relationship between hematological indices and RA disease activity based on DAS28-ESR [10,11,12,14,16]. Unfortunately, these studies did not confirm the association with other disease indices, such as CDAI and SDAI, which are more reliable than DAS28-ESR. This study identified that hematological indices were not related to CDAI and SDAI (data not shown). It is estimated that the differences between DAS28-ESR and CDAI/SDAI in relation to hematological indices are largely due to two factors. First, this study found that ESR showed a higher correlation with hematological indices than CRP. In the case of SDAI containing CRP as a component, it is presumed that the correlation with hematological indices could be insufficient compared to DAS28-ESR containing ESR as a component. Second, there are few studies that have confirmed the relationship between the number of affected joints, such as tender and swollen joint count and hematological indices, although only DAS28-ESR was used for assessment of disease activity in previous studies [10,11,12,14,16]. Interestingly, we observed that hematological indices showed little correlation with the number of affected joints. It seems difficult to provide a reasonable explanation for the relationship between the number of tender joint/swollen joint counts and the hematological indices. Future studies need to clarify these indeterminate relationships.

The present study has several limitations with respect to the interpretation of our results. First, because this study collected and analyzed clinical information, laboratory data, and disease activity parameters retrospectively, it was not possible to confirm the detailed reasons for discontinuation of follow-up and the adverse effects that were the causes of dropout among the recruited patients. Second, hematological indices are usually dependent on counts of whole blood cells. One recent study demonstrated that it takes several weeks to months for changes in lymphocyte, platelet, and neutrophil counts to stabilize [32]. Our ability to confirm the exact trend in changes of hematological parameters, such as lymphocyte, neutrophil, hemoglobin, and platelet counts, was therefore limited because these parameters were measured only at the time of the first administration of JAK inhibitors and then 24 weeks later. Third, the size of the study population enrolled in this study was relatively small. Additional studies with more patients are needed to verify our results.

## 5. Conclusions

This study revealed that hematological indices, SII, NHL score, NLR, and PLR, were significantly correlated with DAS28-ESR in an assessment of disease activity in RA patients treated with JAK inhibitors. However, it is necessary to consider alterations in hematological parameters caused by treatment with JAK inhibitors. Although hematological indices were not related to the number of affected joints, the correlation with DAS28-ESR can be helpful in estimating at least the level of disease activity provided by DAS28-ESR without joint evaluation in clinical practice. With the results obtained, this study is the first to confirm the relevance of hematological indices and composite indices, such as DAS28, to the evaluation of disease activity in patients receiving JAK inhibitors. In addition, it was confirmed that hematological indices were weak predictors of changes in disease activity or treatment response. Therefore, caution is necessary in using hematological indices for assessing disease activity. Through additional prospective longitudinal studies using JAK inhibitors, it will be necessary to verify the usefulness of hematological indices as available disease activity and treatment response markers in patients with RA.

## Figures and Tables

**Figure 1 medicina-58-00426-f001:**
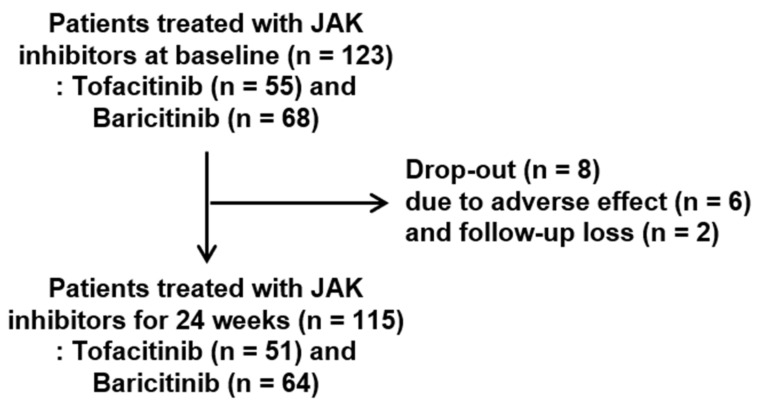
Study population flow chart. Abbreviation: JAK, Janus kinase.

**Figure 2 medicina-58-00426-f002:**
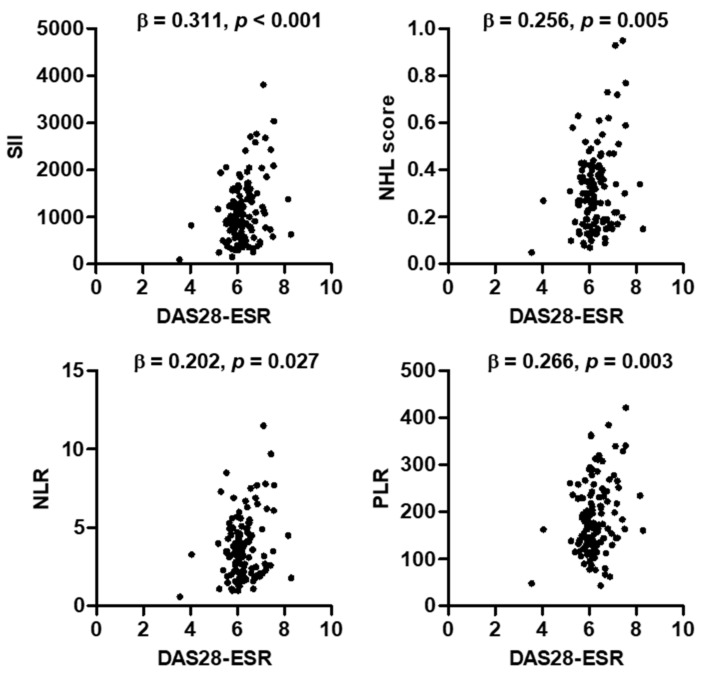
Correlations between DAS28-ESR and hematological indexes at baseline. *p*-values were calculated by multivariate regression analysis after adjustment for age and gender. Abbreviations: DAS, disease activity score; ESR, erythrocyte sedimentation rate; SII, systemic immune-inflammation index; NHL, neutrophil-to-hemoglobin and lymphocyte ratio; NLR, neutrophil-to-lymphocyte ratio; PLR, platelet-to-lymphocyte ratio.

**Table 1 medicina-58-00426-t001:** Baseline characteristics of the study population.

Variables	JAK Inhibitors (*n* = 123)	Healthy Controls (*n* = 80)	*p*-Value
Age (years)	55.6 ± 11.3	58.1 ± 9.8	0.100
Gender (female)	108 (87.8)	75 (93.8)	0.165
Disease duration (months)	189.0 ± 87.4		
Rheumatoid factor (IU/mL)	43.0 (18.0, 118.0)	16.0 (10.0, 57.0)	<0.001
Anti-cyclic citrullinated peptide (U/mL)	276.0 (43.8, 500.0)	8.0 (1.6, 26.0)	<0.001
Disease activity indices			
Erythrocyte sedimentation rate (mm/h)	44.0 (28.0, 59.0)	16.0 (8.3, 30.0)	<0.001
C-reactive protein level (mg/L)	12.1 (5.5, 25.9)	0.7 (0.6, 1.5)	<0.001
Tender joint count	11.0 ± 4.5		
Swollen joint count	7.5 ± 3.6		
Patient VAS (mm)	78.4 ± 9.3		
Physician VAS (mm)	57.1 ± 10.9		
DAS28-ESR	6.2 ± 0.7		
Current medications			
Tofacitinib	55 (44.7)		
Baricitinib	68 (55.3)		
Glucocorticoid	115 (93.5)		
Methotrexate	114 (92.7)		
Hydroxychloroquine	27 (22.0)		
Sulfasalazine	50 (40.7)		
Leflunomide	21 (17.1)		
Tacrolimus	17 (13.8)		

*p*-values were calculated by Student’s *t*-test, Mann–Whiney U test, or chi-squared test. Data were described as numbers of cases (%) for qualitative variables and as means (standard deviation, SD) or medians (interquartile range) for quantitative variables. Abbreviations: DAS, disease activity score; ESR, erythrocyte sedimentation rate; CRP, C-reactive protein level; VAS, visual analogue scale; SII, systemic immune-inflammation index; NHL, neutrophil-to-hemoglobin and lymphocyte ratio; NLR, neutrophil-to-lymphocyte ratio; PLR, platelet-to-lymphocyte ratio.

**Table 2 medicina-58-00426-t002:** Comparison of hematological indices between JAK inhibitors and controls.

Variables	JAK Inhibitors (*n* = 123)	Healthy Controls (*n* = 80)	*p*-Value
Whole blood cell count			
White blood cell (×10^3^/µL)	8.0 ± 1.9	5.4 ± 1.3	<0.001
Hemoglobin (g/dL)	12.2 ± 1.2	12.8 ± 1.2	<0.001
Platelet (×10^3^/µL)	287.3 ± 80.1	239.9 ± 65.1	<0.001
Neutrophil (%)	68.0 ± 10.0	54.1 ± 10.7	<0.001
Lymphocyte (%)	22.0 ± 8.6	35.4 ± 9.9	<0.001
Neutrophil (×10^3^/µL)	5.6 ± 1.9	3.0 ± 1.0	<0.001
Lymphocyte (×10^3^/µL)	1.7 ± 0.6	1.9 ±0.6	0.027
Hematological indices			
SII	928.7 (587.9, 1385.6)	380.5 (236.6, 543.6)	<0.001
NHL score	0.281 (0.179, 0.400)	0.121 (0.087, 0.169)	<0.001
NLR	3.431 (2.201, 4.736)	1.566 (1.125, 2.146)	<0.001
PLR	174.6 (133.6, 235.0)	130.8 (101.6, 164.0)	<0.001

*p*-values were calculated by Student’s *t*-test or Mann–Whiney U test. Data were described as means (standard deviation) or medians (interquartile range) for quantitative variables. Abbreviations: SII, systemic immune-inflammation index; NHL, neutrophil-to-hemoglobin and lymphocyte ratio; NLR, neutrophil-to-lymphocyte ratio; PLR, platelet-to-lymphocyte ratio.

**Table 3 medicina-58-00426-t003:** Correlation between changes in hematological indexes and disease activity parameters in rheumatoid arthritis (*n* = 115).

Change of Disease Activity Parameters	Change of Hematological Indexes							
	ΔSII		ΔNHL Score		ΔNLR		ΔPLR	
	*r*	*p*	*r*	*p*	*r*	*p*	*r*	*p*
ΔDAS28-ESR	0.233	0.012	0.221	0.018	0.199	0.033	0.244	0.009
ΔESR	0.383	0.000	0.312	0.001	0.247	0.008	0.427	0.000
ΔCRP	0.208	0.026	0.139	0.139	0.099	0.293	0.313	0.001
ΔTender joint count	0.007	0.944	0.004	0.970	−0.016	0.861	0.035	0.708
ΔSwollen joint count	0.079	0.402	0.123	0.191	0.084	0.374	0.046	0.629
ΔPatient VAS	0.066	0.484	0.030	0.749	0.037	0.694	0.140	0.137
ΔPhysician VAS	0.201	0.130	0.149	0.266	0.170	0.201	0.272	0.039

*p*-values were calculated by Pearson correlation analysis. Data were described as correlation coefficients (*r*). Abbreviations: DAS, disease activity score; ESR, erythrocyte sedimentation rate; CRP, C-reactive protein level; VAS, visual analogue scale; SII, systemic immune-inflammation index; NHL, neutrophil-to-hemoglobin and lymphocyte ratio; NLR, neutrophil-to-lymphocyte ratio; PLR, platelet-to-lymphocyte ratio.

**Table 4 medicina-58-00426-t004:** Comparison of changes in disease activity markers before and after use of JAK inhibitor treatment (*n* = 115).

Variables	Baseline	After 24 Weeks	*p*-Value
Hematological indices			
SII	915.1 (576.3, 1378.7)	697.9 (439.3, 1031.8)	<0.001
NHL score	0.281 (0.175, 0.394)	0.197 (0.137, 0.279)	<0.001
NLR	3.419 (2.179, 4.736)	2.393 (1.782, 3.313)	<0.001
PLR	172.6 (132.6, 235.0)	155.5 (125.8, 217.1)	0.044
Disease activity parameters			
DAS28-ESR	6.2 ± 0.7	4.3 ± 0.7	<0.001
ESR	46.5 ± 24.5	27.3 ± 20.9	<0.001
CRP	18.2 ± 18.8	4.3 ± 7.6	<0.001
Whole blood cell count			
White blood count (×10^3^/L)	8.0 ± 1.9	7.0 ± 2.1	<0.001
Hemoglobin (g/dL)	12.2 ± 1.2	12.4 ± 1.3	0.044
Neutrophil (%)	67.8 ± 1.1	62.8 ± 10.2	<0.001
Lymphocyte (%)	22.3 ± 8.7	27.6 ± 9.4	<0.001
Neutrophil (×10^3^/L)	5.5 ± 1.8	4.4 ± 1.8	<0.001
Lymphocyte (×10^3^/L)	1.7 ± 0.6	1.8 ± 0.7	0.007

*p*-values were calculated by means of a paired samples *t*-test or the Wilcoxon signed-rank test. Abbreviations: DAS, disease activity score; ESR, erythrocyte sedimentation rate; CRP, C-reactive protein; SII, systemic immune-inflammation index; NHL, neutrophil-to-hemoglobin and lymphocyte ratio; NLR, neutrophil-to-lymphocyte ratio; PLR, platelet-to-lymphocyte ratio.

**Table 5 medicina-58-00426-t005:** Analysis for non-response to JAK inhibitors for 24 weeks in patients with RA.

	Univariate Model		Multivariate Model 1		Multivariate Model 2	
Variable at Baseline	OR (95% CI)	*p*-Value	OR (95% CI)	*p*-Value	OR (95% CI)	*p*-Value
Swollen joint count	1.157 (1.009–1.328)	0.037	1.189 (1.027–1.378)	0.021	1.262 (1.038–1.535)	0.020
Rheumatoid factor	1.002 (1.000–1.005)	0.042	1.002 (1.000–1.004)	0.076		
C-reactive protein	1.030 (1.003–1.057)	0.026	1.028 (1.000–1.057)	0.052		
SII	1.000 (1.000–1.001)	0.225				
NHL score	8.793 (0.378–204.795)	0.176				
NLR	1.212 (0.912–1.811)	0.186				
PLR	1.001 (0.993–1.008)	0.888				

Data were described as odds ratios (ORs) and 95% confidence intervals (CIs). *p*-values were obtained by univariate and multivariate logistic regression analysis. Model 1, adjusted with age and gender; model 2, adjusted with age, gender, disease duration, and DAS28-ESR. Abbreviations: SII, systemic immune-inflammation index; NHL, neutrophil-to-hemoglobin and lymphocyte ratio; NLR, neutrophil-to-lymphocyte ratio; PLR, platelet-to-lymphocyte ratio.

## Data Availability

The data underlying this article will be shared on reasonable request by the corresponding author.

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
