# Peer review of "Association between Hematological Indicesand Disease Activity in Patients with Rheumatoid Arthritis Treated with Janus Kinase Inhibitors for 24 Weeks"

_medicina, 2022, doi:10.3390/medicina58030426_

Round 1

Reviewer 1 Report

Reviewer comments

Abstract

The conclusion does not cope with the results

Line 57,58,59 ..The language need to be refined

Line 67 ..many criteria for RA have been released ..why the authors chose this old one

Line 71..Tacrolimus not commonly used in RA ..the authors have not mentioned other more common DMARDs

Line 78 ..the study is retrospectively ..how the authors collected data at basline

Materials and methods ..

The authors have not mentioned the ethical approval or consents obtained from the patients

Results

P values were calculated by Student t-test or chi-square test. . Data were described as number of case (%) for qualitative variables or mean (stand-ard deviation, SD) for quantitative variables….this part is usually written in statistical analysis ..not here  

Abbreviation:.. should not be written here

Table 1 .. 122.2 ± 194.6 ..this description is not correct

Also this ,269.6 ± 208.5

This is not correct .. 46.2 ± 24.2

Why the authors did not compare between patients on JAK and patients not on JAK

Author Response

Dear Editor

Manuscript ID: medicina-1612364

Title: Hematological indices are potential markers for disease activity in patients with rheumatoid arthritis treated with Janus kinase inhibitors for 24 weeks

   Thank for the editor and reviewers of the ‘Medicina’ for reviewing our manuscript. We have made some corrections and clarifications in the revised manuscript according to the editor’s or reviewer's comments. You can find out tracing marks for changes in revised manuscript. The changes are summarized below:

Referee: 1

Abstract

  1. The conclusion does not cope with the results

Answer)

Thanks for your valuable comment. The results and conclusions of this study might be completely in agreement with some aspects. Since we also agree that hematological indices have a limited role as useful marker, the description of the marker has been deleted.

  1. Line 57,58,59 ..The language need to be refined

Answer)

Thanks for your kind comment. The sentence “There has not been a study evaluating …” at line from 57 to 59 is changed into “No studies have evaluated whether …”.

  1. Line 67 ..many criteria for RA have been released ..why the authors chose this old one

Answer)

Thanks for your thoughtful comment. It is a mistake to state that we used older the 1987 diagnostic criteria for RA. The classification criteria for rheumatoid arthritis patients enrolled in this study was the criteria presented by ACR/EULAR in 2010. The old reference is changed into new reference, as follows “17. Aletaha D.; Neogi T.; Silman A.J.; Funovits J.; Felson D.T.; Bingham C.O.; Birnbaum N.S.; Burmester G.R.; Bykerk V.P.; Cohen M.D.; Combe B.; Costenbader K.H.; Dougados M.; Emery P.; Ferraccioli G.; Hazes J.M.; Hobbs K.; Huizinga T.W.; Kavanaugh A.; Kay J.; Kvien T.K.; Laing T.; Mease P.; Ménard H.A.; Moreland L.W.; Naden R.L.; Pincus T.; Smolen J.S.; Stanislawska-Biernat E.; Symmons D.; Tak P.P.; Upchurch K.S.; Vencovský J.; Wolfe F.; Hawker G. 2010 Rheumatoid arthritis classification criteria: an American College of Rheumatology/European League Against Rheumatism collaborative initiative. Arthritis. Rheum. 2010, 62, 2569-2581.”.

  1. Line 71..Tacrolimus not commonly used in RA ..the authors have not mentioned other more common DMARDs

Answer)

Thanks for your kind comment. In Korea, tacrolimus is widely used in clinical practice of rheumatic clinics in place of cyclosporin A. Other commonly used DMARDs including sulfasalazine, hydroxychloroquine, and azathioprine are added in the same sentence.

  1. Line 78 ..the study is retrospectively ..how the authors collected data at baseline

Answer)

Thanks for your kind comment. Clinical information was obtained from review of medical records and individual interview with participants. Thus, we revise the sentence at line 78 as follows “Clinical information was collected from patients with RA through review of medical records and individual interview with participants at the time of initial JAK inhibitor ad-ministration”.

Materials and methods ..

The authors have not mentioned the ethical approval or consents obtained from the patients

Answer)

Thanks for your comment. We have already described ethical issues in the section “Institutional Review Board Statement” at the end of the manuscript like this; “Institutional Review Board Statement: This study was performed according to the ethical standards of the institutional and/or national research committee and with the 1964 Helsinki declaration and its later amendments or comparable ethical standards. This study was approved by the Institutional Review Board at Daegu Catholic University Medical Center (CR-21-166)”.

Results

  1. P values were calculated by Student t-test or chi-square test. Data were described as number of case (%) for qualitative variables or mean (standard deviation, SD) for quantitative variables….this part is usually written in statistical analysis ..not here. Abbreviation:.. should not be written here

Answer)

Thanks for your comment. The part mentioned by the reviewer was originally described as a footnote in Table 1, but it seems to have been placed on Table 1 in the editing process of MDPI. We move this part back to the footnote of Table 1.

  1. Table 1 .. 122.2 ± 194.6, 269.6 ± 208.5, and 46.2 ± 24.2 ..these description is not correct

Answer)

Thanks for your comment. We confirm that these data are correct, even though we check the results of this statistical analysis again.

  1. Why the authors did not compare between patients on JAK and patients not on JAK

Answer)

Thanks for your valuable comment. The main purpose of our study was limited to analyze the usefulness of JAK inhibitors on the activity change of rheumatoid arthritis. Previous studies already analyzed the effect of each biologic DMARD on clinical values of hematological indices. In this study, the relationship between the hematological effects of JAK inhibitors and disease activity in patients treated with JAK inhibitors has already been discussed in the discussion section.

Reviewer 2 Report

Good article, very interesting. For my part, I have no revisions to make.

Author Response

Dear Editor

Manuscript ID: medicina-1612364

Title: Hematological indices are potential markers for disease activity in patients with rheumatoid arthritis treated with Janus kinase inhibitors for 24 weeks

   Thank for the editor and reviewers of the ‘Medicina’ for reviewing our manuscript. We have made some corrections and clarifications in the revised manuscript according to the editor’s or reviewer's comments. You can find out tracing marks for changes in revised manuscript. The changes are summarized below:

Referee: 2

Good article, very interesting. For my part, I have no revisions to make.

Answer)

We thank the reviewers for their thoughtful and undeserved comments.

Reviewer 3 Report

1) Introduction. L29-31. Rheumatoid arthritis (RA) is a systemic autoimmune disease characterized by enhanced synovial inflammation, neovascularization, and osteoclast formation, causing inflammatory arthritis and progressive joint damage that leads to deterioration of quality of life and increased functional disability [1, 2]. Please improve this paragraph and add these references:

  1. A) Figus, F. A., Piga, M., Azzolin, I., McConnell, R., & Iagnocco, A. (2021). Rheumatoid arthritis: Extra-articular manifestations and comorbidities. Autoimmunity reviews20(4), 102776. https://doi.org/10.1016/j.autrev.2021.102776
  2. B) Ruaro, B., Casabella, A., Paolino, S., Pizzorni, C., Ghio, M., Seriolo, C., Molfetta, L., Odetti, P., Smith, V., & Cutolo, M. (2018). Dickkopf-1 (Dkk-1) serum levels in systemic sclerosis and rheumatoid arthritis patients: correlation with the Trabecular Bone Score (TBS). Clinical rheumatology37(11), 3057–3062. https://doi.org/10.1007/s10067-018-4322-9

2) Introduction. L 59-61. The purpose of this study is to investigate whether hematological indices such as NLR and PLR reflect changes of disease activity in patients with RA treated with JAK inhibitors including tofacitinib and baricitinib for 24 weeks.

3) Could you please clarify any different regarding treatment regimes?

4) Table 2. Comparison of hematological indices between JAK inhibitors and controls. Please add the r-values.

5) Discussion. JAK inhibitors, small molecular compounds, and a novel class of oral tsDMARDs are characterized by blockage of intracellular signaling of the JAK-STAT pathway, which plays a crucial role in regulation of the inflammatory response in inflammatory or immune cells, unlike bDMARDs, which bind to extracellular target molecules or inhibit their activity. Nowadays, JAK inhibitors are anti-rheumatic drugs that have become an option for treatment of RA [15, 16]. Please summarise here the most important results of the study.

6) The present study has several limitations to the interpretation of our results. First, because this study collected and analyzed clinical information, laboratory data, and disease activity parameters retrospectively, it was insufficient to confirm the detailed reasons for discontinuation of follow-up and adverse effects that were causes of dropout among the recruited patients. Second, hematological indices usually are dependent on count of whole blood cells. One recent study demonstrated that it takes several weeks to months for the changes in lymphocyte, platelet, and neutrophil counts to stabilize [28, 30] and was limited to confirm the exact trend in changes of hematological parameters such as lymphocyte, neutrophil, hemoglobin, and platelet counts because these parameters were measured only at the time of the first administration of JAK inhibitors and then 24 weeks later. Please underline the small size of study.

7) Conclusion This study revealed that hematological indices SII, NHL score, NLR, and PLR were significantly correlated with DAS28-ESR in the assessment of disease activity of RA patients treated with JAK inhibitors. However, it is necessary to consider alterations in hematological parameters caused by treatment with JAK inhibitors. Although hematological indices were not related to the number of affected joints, the correlation with DAS28-ESR can be helpful in estimating at least the level of disease activity provided by DAS28-ESR without joint evaluation in clinical practice. Please underline the novelty of the study and the clinical implication of these observations.

Author Response

Dear Editor

Manuscript ID: medicina-1612364

Title: Hematological indices are potential markers for disease activity in patients with rheumatoid arthritis treated with Janus kinase inhibitors for 24 weeks

   Thank for the editor and reviewers of the ‘Medicina’ for reviewing our manuscript. We have made some corrections and clarifications in the revised manuscript according to the editor’s or reviewer's comments. You can find out tracing marks for changes in revised manuscript. The changes are summarized below:

Referee: 3

1) Introduction. L29-31. Rheumatoid arthritis (RA) is a systemic autoimmune disease characterized by enhanced synovial inflammation, neovascularization, and osteoclast formation, causing inflammatory arthritis and progressive joint damage that leads to deterioration of quality of life and increased functional disability [1, 2]. Please improve this paragraph and add these references:

Answer)

Thanks for your valuable comment. As recommended by the reviewer, we have added the two references [3, 4] listed below.

  1. Figus F.A.; Piga M.; Azzolin I.; McConnell R.; Iagnocco A. Rheumatoid arthritis: Extra-articular manifestations and comorbidities. Autoimmun. Rev. 2021, 20, 102776.
  2. Ruaro B.; Casabella A.; Paolino S.; Pizzorni C.; Ghio M.; Seriolo C.; Molfetta L.; Odetti P.; Smith V.; Cutolo M. Dickkopf-1 (Dkk-1) serum levels in systemic sclerosis and rheumatoid arthritis patients: correlation with the Trabecular Bone Score (TBS). Clin. Rheumatol. 2018, 37, 3057-3062.

2) Introduction. L 59-61. The purpose of this study is to investigate whether hematological indices such as NLR and PLR reflect changes of disease activity in patients with RA treated with JAK inhibitors including tofacitinib and baricitinib for 24 weeks.

Answer)

Thanks for your comment. We are so sorry that No. 2 mentioned by the reviewer cannot provide an answer because there was no clear question.

3) Could you please clarify any different regarding treatment regimens?

Answer)

Thanks for your valuable comment. No. 3 mentioned by the reviewer does not have a detailed description of the question. Presumably, it is judged as a question for question 2, so I will answer as follows. We add detailed description about treatment regimens as follows “Small molecule JAK inhibitors are oral targeted DMARDs, characterized by the blockade of intracellular JAK-STAT pathway mediated by multiple cytokines involved in the immune-mediated inflammatory response in RA, unlike bDMARDs that regulate inflammatory cytokines, T-lymphocytes or B-lymphocytes”.

4) Table 2. Comparison of hematological indices between JAK inhibitors and controls. Please add the r-values.

Answer)

Thanks for your comment. Do the r values mentioned by the reviewer mean correlation coefficients (r) that confirm the correlation between the two groups? We have already presented the p values because it is a comparison between the two groups.

5) Discussion. JAK inhibitors, small molecular compounds, and a novel class of oral tsDMARDs are characterized by blockage of intracellular signaling of the JAK-STAT pathway, which plays a crucial role in regulation of the inflammatory response in inflammatory or immune cells, unlike bDMARDs, which bind to extracellular target molecules or inhibit their activity. Nowadays, JAK inhibitors are anti-rheumatic drugs that have become an option for treatment of RA [15, 16]. Please summarize here the most important results of the study.

Answer)

Thanks for your kind comment. Parts that overlap with those described in the introduction are deleted. At the end of the first paragraph of the discussion, I would like to point out that the main results of this study have already been described.

6) The present study has several limitations to the interpretation of our results. First, because this study collected and analyzed clinical information, laboratory data, and disease activity parameters retrospectively, it was insufficient to confirm the detailed reasons for discontinuation of follow-up and adverse effects that were causes of dropout among the recruited patients. Second, hematological indices usually are dependent on count of whole blood cells. One recent study demonstrated that it takes several weeks to months for the changes in lymphocyte, platelet, and neutrophil counts to stabilize [28, 30] and was limited to confirm the exact trend in changes of hematological parameters such as lymphocyte, neutrophil, hemoglobin, and platelet counts because these parameters were measured only at the time of the first administration of JAK inhibitors and then 24 weeks later. Please underline the small size of study.

Answer)

Thanks for your valuable comment. The limitation that size of this study population is small is added as follows “Third, the size of study population enrolled in this study was relatively small. Additional studies with more patients are needed to verify the results of our study”.

7) Conclusion This study revealed that hematological indices SII, NHL score, NLR, and PLR were significantly correlated with DAS28-ESR in the assessment of disease activity of RA patients treated with JAK inhibitors. However, it is necessary to consider alterations in hematological parameters caused by treatment with JAK inhibitors. Although hematological indices were not related to the number of affected joints, the correlation with DAS28-ESR can be helpful in estimating at least the level of disease activity provided by DAS28-ESR without joint evaluation in clinical practice. Please underline the novelty of the study and the clinical implication of these observations.

Answer)

Thanks for your valuable comment. We add the novelty of this study and clinical implication as follows “Based on the results of this study, this study is the first study to confirm the relevance of a hematological indices with composite indices such as DAS28 that evaluates the disease activity of patients receiving JAK inhibitors”.

Round 2

Reviewer 1 Report

  1. Table 1 .. 122.2 ± 194.6, 269.6 ± 208.5, and 46.2 ± 24.2 ..these description is not correct..It is wrong that double the standard deviation is higher than the mean ..this means that we have negative values ..so ,something wrong in these results  

Author Response

  1. Table 1 .. 122.2 ± 194.6, 269.6 ± 208.5, and 46.2 ± 24.2 ..these description is not correct

Answer)

Thanks for your valuable comment. We checked whether these variables have a normal distribution, and confirmed that they show a non-normal distribution. Thus, we revised the results for these variables again as median values (interquartile range) in table 1, and there are no changes in statistical significance.